# Influence of Lateral Translation of Lowest Instrumented Vertebra on L4 Tilt and Coronal Balance for Thoracolumbar and Lumbar Curves in Adolescent Idiopathic Scoliosis

**DOI:** 10.3390/jcm12041389

**Published:** 2023-02-09

**Authors:** Katsuhisa Yamada, Hideki Sudo, Yuichiro Abe, Terufumi Kokabu, Hiroyuki Tachi, Tsutomu Endo, Takashi Ohnishi, Daisuke Ukeba, Katsuro Ura, Masahiko Takahata, Norimasa Iwasaki

**Affiliations:** 1Department of Orthopaedic Surgery, Hokkaido University Hospital, Sapporo 060-8638, Japan; 2Department of Advanced Medicine for Spine and Spinal Cord Disorders, Faculty of Medicine and Graduate School of Medicine, Hokkaido University, Sapporo 060-8638, Japan; 3Department of Orthopaedic Surgery, Eniwa Hospital, Eniwa 061-1449, Japan

**Keywords:** adolescent idiopathic scoliosis, thoracolumbar/lumbar curve, lowest instrumented vertebra, LIV translation, anterior spinal fusion, posterior spinal fusion

## Abstract

This study aimed to evaluate the lowest instrumented vertebra translation (LIV-T) in the surgical treatment of thoracolumbar/lumbar adolescent idiopathic scoliosis and to analyze the radiographic parameters in relation to LIV-T and L4 tilt and global coronal balance. A total of 62 patients underwent posterior spinal fusion (PSF, *n* = 32) or anterior spinal fusion (ASF, *n* = 30) and were followed up for a minimum of 2 years. The mean preoperative LIV-T was significantly larger in the ASF group than the PSF (*p* < 0.01), while the final LIV-T was equivalent. LIV-T at the final follow-up was significantly correlated with L4 tilt and the global coronal balance (r = 0.69, *p* < 0.01, r = 0.38, *p* < 0.01, respectively). Receiver-operating characteristic analysis for good outcomes, with L4 tilt <8° and coronal balance <15 mm at the final follow-up, calculated the cutoff value of the final LIV-T as 12 mm. The cutoff value of preoperative LIV-T that would result in the LIV-T of ≤12 mm at the final follow-up was 32 mm in PSF, although no significant cutoff value was calculated in ASF. ASF can centralize the LIV better than PSF with a shorter segment fusion, and could be useful in obtaining a good curve correction and global balance without fixation to L4 in cases with large preoperative LIV-T.

## 1. Introduction

The goal of adolescent idiopathic scoliosis (AIS) surgical treatment is to maximally correct the spinal deformity in three anatomical planes [1,2]. Patients with a Lenke type 5 AIS curve, which is characterized by a major thoracolumbar/lumbar (TL/L) curve with a compensatory nonstructural thoracic curve [3], are often reported with coronal imbalance [4,5], and obtaining good global balance forms an important part of treatment. A favorable correction of the L4 tilt is also known to be critical in preventing the progression of the uninstrumented lumbar curve, lumbar disc degeneration, and low back pain [6,7,8]. Lenke 5 curve treatment planning should, thus, consider the TL/L curve correction, as well as the global spinal balance and L4 tilt.

There has been controversy regarding the lowest instrumented vertebra (LIV) for the corrective surgery of the TL/L curve [9]. LIV selection is important in preserving lumbar spine mobility while providing optimal correction, and inappropriate LIV selection can worsen the unfused curve and cause distal adding-on and spinal imbalance [4,9]. Since there have been some reports of low back pain progression and loss of the lumbar range of motion when L4 is included in the fusion segment [10], L3 is often selected as the LIV to maximize the lumbar motion segments and minimize future disc degeneration [11,12,13,14,15,16]. It has also been reported that the LIV translation and inclination should be considered when selecting the LIV [6,14,17]. Leveling and centering the LIV is expected to restore the vertebrae to their anatomically normal position, correct the lumbar curve, and provide a better global balance of the spine [6,14,17]. However, there are limited data on the factors associated with favorable outcomes for LIV selection [14]. Furthermore, while both the anterior spinal fusion (ASF) and posterior spinal fusion (PSF) approaches have been found to be safe and effective treatments for Lenke 5 AIS [11,18], there is no consensus on the LIV selection process for each surgical procedure.

We focused on the effect of LIV translation from the center sacral vertical line (CSVL) on the L4 tilt and coronal balance. Determining the relationship between preoperative LIV translation and postoperative outcomes could be important for surgical strategy in terms of LIV and surgical approach selections. The purpose of this study was to evaluate LIV translation for ASF and PSF in the treatment of Lenke 5 AIS and to analyze the effect of LIV translation on radiographic parameters for each surgical procedure.

## 2. Materials and Methods

### 2.1. Subject

After institutional review board approval (approval number: 020-0416), we retrospectively reviewed 62 cases of patients who underwent spinal fusion for Lenke 5 AIS at our institution. We included thirty-two patients (two males, thirty females, follow-up > 2 years) who underwent PSF between 2010 and 2020, and thirty patients (four males, twenty-six females, follow-up > 12 years) who underwent ASF between 1989 and 2000 [11,12]. The outcomes of each surgical approach were then compared and evaluated. Syndromic, neuro-muscular, and congenital scoliosis were excluded. No cases were lost to the follow-up. At the time of surgery, the average age was 15.1 years (11–19 years) and the average Risser grade was 3.6 (0–5).

### 2.2. Radiographic Parameters

Multiple parameters were evaluated using preoperative, postoperative, and final follow-up standing long-cassette posteroanterior and lateral radiographs. The coronal curve measurements of the main thoracic and TL/L curves were obtained. The flexibility of the curve was assessed using the preoperative supine-bending radiographs. To ensure consistency for statistical comparisons, the end vertebrae levels were determined using the preoperative radiographs and measured on subsequent radiographs [11]. The sagittal measurements included thoracic kyphosis (T5-12) and lumbar lordosis (L1-S1). We then measured the lateral displacement of the C7 plumb line from the CSVL to evaluate the global coronal balance, which was defined as a C7 translation from the CSVL (C7-CSVL) [19]. For regional alignment, the main thoracic apical vertebral translation was measured as the distance between the geometric center of the apical vertebra and the C7 plumb line [20]. L4 tilt was measured as the inclination angle of the superior endplate of the vertebra from the horizontal plane. Sagittal balance was evaluated by measuring the absolute displacement value of the C7 plumb line relative to the S1 posterior superior corner as the sagittal vertical axis [21].

The radiographic parameters relevant to the LIV were measured as follows: LIV translation was measured as the distance between the geometric center of the LIV and the C7 plumb line [11]. LIV tilt was defined as the inferior endplate inclination angle of the LIV from the horizontal line. The coronal disc angle immediately below the LIV was considered the LIV disc angle. The LIV tilt and LIV disc angles were defined as negative when they opened to the preoperative convex side, and positive when they opened to the preoperative concave side, as previously reported [20,22].

### 2.3. Surgical Techniques

#### 2.3.1. Posterior Spinal Fusion

Fusion-level selection was based on both standing and bending radiographs. Fusion levels were typically planned from end-to-end vertebrae. However, L3 was selected as the LIV even when L4 was the lower-end vertebra. When the upper-end vertebra was rotated (Nash–Moe grade [23] > 2), the vertebra one or two levels above the upper-end vertebra was selected as the upper instrumented vertebra.

PSF was performed according to the previously reported posterior corrective surgery protocol for the main thoracic curve [1,6,24,25] (Figure 1). Briefly, after posterior exposure, side-loading polyaxial pedicle screws were placed bilaterally at all possible vertebrae. After all-level facetectomy, two rods were connected to the screw heads and simultaneously rotated [1,6,25]. Compression force was segmentally applied between the screws on the convex side to create lumbar lordosis, followed by a distraction force on the concave side. Local bone grafting followed the decortication of the laminae. None of the patients required a brace [1,6,25].

#### 2.3.2. Anterior Spinal Fusion

Fusion levels were typically planned from end-to-end vertebrae, and the horizontalized vertebra without substantial rotation on bending radiographs was selected as the LIV [11,26]. A short fusion was performed in cases with high flexibility of the TL/L curves and/or neutral vertebrae located one level above the lower-end vertebra [11,12,20,27]. The upper instrumented vertebra was determined according to the flexibility of the thoracic compensatory curve, and one segment below the upper-end vertebra was selected for rigid curves [11,12].

Anterior dual-rod instrumentation surgery for the TL/L curve was performed according to previously reported protocols [11,12,26,28] (Figure 1). Briefly, after vertebral exposure via extrapleural retroperitoneal approaches, intervertebral disc and cartilage-end plates were removed. Two screws were inserted through a vertebral staple, and two rods were placed and rotated. Compression force was applied between the screws to correct the scoliosis. An autogenous rib graft was placed at each discectomy site.

(Posterior spinal fusion) A thirteen-year-old girl with a preoperative thoracolumbar (TL) curve of 48° improved to 8° 2 years after surgery. Lower instrumented vertebra translation from the central sacral vertical line (LIV translation) was 32 mm preoperatively and improved to 16 mm at the final follow up.(Anterior spinal fusion) A fourteen-year-old girl with a preoperative TL curve of 53° improved to 10° 18 years after surgery. LIV translation improved from 49 mm preoperatively to 11 mm at the final follow-up.

Regarding the choice of ASF or PSF for thoracolumbar AIS, ASF provides better correction with fewer fixed spinal segments than PSF because of the increased apical mobility induced by anterior discectomies [11,22]. Meanwhile, we have reported that the anterior approach for thoracic AIS may decrease respiratory function in the long term, and that thoracotomy may affect the pulmonary function data [27]. When the upper instrumented vertebra is more cephalad than T10, ASF for thoracolumbar AIS would require a thoracotomy, which may affect respiratory function. For these reasons, our current surgical strategy for thoracolumbar AIS has been PSF for cases with upper-end vertebra cephalad to T10, and ASF for cases with UEV caudal to T11.

### 2.4. Data Analysis

We compared demographic data and radiographic parameters between patients who underwent anterior and posterior surgery. Surgical outcomes were evaluated in terms of L4 tilt and global coronal balance, and, based on previous reports, good outcomes were defined as cases with an L4 tilt < 8° and C7-CSVL < 15 mm at the final follow-up [8,11,17,29]. The association between the LIV translation and surgical outcome was statistically evaluated.

All data are expressed as the mean ± standard deviation. Independent sample *t* tests were used to compare between-group differences in demographic data and radiographic parameters. A Pearson’s correlation coefficient test was used to assess the relationships between continuous data. A receiver-operating characteristic (ROC) analysis was conducted to assess the cutoff point of the LIV translation. Statistical significance was defined as *p* < 0.05.

## 3. Results

### 3.1. Patient Demographic Data

The demographic data of the 62 patients are shown in Table 1. The mean age at surgery was significantly younger in the ASF group than the PSF group (*p* < 0.01), and the Risser grade was also lower in the ASF group (*p* < 0.01). Instrumentation length was significantly longer in the PSF group than the ASF group (*p* < 0.01), and the LIV ranged from L2 to L3 in the ASF group, while the LIV was L3 in all patients of the PSF group. Patients who underwent PSF had significantly shorter surgical times than those who underwent ASF (*p* < 0.01), but there was no significant difference in intraoperative blood loss between the two groups (Table 1).

### 3.2. Radiographic Parameters

#### 3.2.1. Radiographic Parameters for All 62 Cases

Coronal and sagittal radiographic parameters for all 62 cases are summarized in Table 2. The average TL/L curve before surgery was 51°, with a significant postoperative improvement to 10° (*p* < 0.01). The average compensatory thoracic curve before surgery was 29°, which improved significantly to 16° at the final follow-up (*p* < 0.01). In the sagittal plane, the thoracic kyphosis angle was significantly larger at the final follow-up than preoperatively (*p* < 0.01). Regarding global trunk balance, the average translation of the C7-CSVL was 22 mm preoperatively with no immediate postoperative improvement (*p* = 0.10); however, significant improvement to 10 mm was seen at the final follow-up (*p* < 0.01). There was no significant difference in the sagittal vertical axis among the preoperative, postoperative, and final measurements (*p* = 0.43, *p* = 0.12, respectively). As for regional alignment, the apical vertebral translation significantly improved from 52 mm to 10 mm at the final follow-up (*p* < 0.01). The L4 tilt showed significant improvement from 22° preoperatively to 6° at the final follow-up (*p* < 0.01). The mean LIV translation was 33 mm preoperatively, significantly improving to 12 mm at the final follow-up (*p* < 0.01). The LIV tilt significantly improved from 26° preoperatively to −2° at the final follow-up (*p* < 0.01). There was a significant change in the preoperative disc angle below the LIV and the final follow-up (*p* < 0.01).

#### 3.2.2. Radiographic Parameters of ASF and PSF Group

The radiological parameters for each of the ASF and the PSF surgical procedures are shown in Table 3. The preoperative TL/L curves were significantly larger in the ASF group than the PSF group (*p* < 0.01), but were equivalent at the final follow-up (*p* = 0.14). The thoracic curve correction rates were equivalent between the two groups (*p* = 0.18). Thoracic kyphosis at each time point and lumbar lordosis after surgery were significantly larger in the ASF group than the PSF group. The preoperative C7-CSVL, L4 tilt, and LIV translation were significantly larger in the PSF group than the ASF group (*p* = 0.02, *p* = 0.04, *p* < 0.01, respectively) and improved equivalently at the final follow-up (*p* = 0.32, *p* = 0.12, *p* = 0.39, respectively). While the preoperative LIV tilt was equivalent between the two groups (*p* = 0.24), it was significantly larger in the PSF group than the ASF group at follow-up (*p* < 0.01).

#### 3.2.3. Correlation Analysis

Correlation analysis was conducted between the LIV translation, which was obtained from CSVL values, and the radiographic parameters at the final follow-up (Table 4). The LIV translation was significantly correlated with the L4 tilt and C7-CSVL (r = 0.69, *p* < 0.01; r = 0.38, *p* < 0.01, respectively). Similarly, the LIV translation was significantly correlated with the TL/L curve correction rate and correction loss (r = −0.64, *p* < 0.01; r = 0.42, *p* < 0.01, respectively), and also significantly correlated with the apical vertebral translation and the LIV tilt (r = 0.83, *p* < 0.01; r = 0.69, *p* < 0.01, respectively). Conversely, no correlation was found between the LIV translation and the disc angle below the LIV (*p* = 0.10).

#### 3.2.4. Subgroup Analysis

Surgical outcomes were evaluated in terms of L4 tilt and global coronal balance, and 38 cases with L4 tilt < 8° and C7-CSVL < 15 mm at final follow-up were defined as good outcomes. Radiographic parameters at the final follow-up were compared between the 38 good-outcome cases and the remaining 24 cases (Table 5). Both groups were similar at baseline for the following parameters: age, Risser sign, number of vertebrae in major curve, surgical procedure (ASF, PSF), and instrumentation length. The L4 tilt < 8° and C7-CSVL < 15 mm group showed significantly lower TL/L curves, apical vertebral translation, sagittal vertical axis, and LIV translation than the L4 tilt ≥ 8° or C7-CSVL ≥ 15 mm group, (Table 5). No significant differences were found between the groups in thoracic curve, thoracic kyphosis, lumbar lordosis, and disc angle below the LIV.

ROC analysis established the cutoff point for good outcomes of LIV translation at the final follow-up as 12 mm, with a sensitivity of 75.0% and specificity of 74.3% (*p* < 0.01, the area under the ROC curve [AUC]: 0.82, 95% confidence interval [CI]: 0.71–0.94) (Figure 2). Thus, the preoperative LIV translation cutoff value that would result in the LIV translation of 12 mm or less at the final follow-up was determined by a ROC analysis for each surgical approach. In the PSF group, the preoperative LIV translation cutoff point was 32 mm, with a sensitivity of 76.9% and specificity of 84.2% (*p* < 0.01, AUC: 0.81, 95% CI: 0.65–0.97) (Figure 3). Furthermore, no significant preoperative LIV translation cutoff values were obtained in the ASF group (*p* = 0.27) (Figure 3).

Based on a 32 mm preoperative LIV translation cutoff point in the PSF group, we analyzed 32 cases with the preoperative LIV translation of ≥32 mm (19 ASF and 13 PSF cases). The demographic data of these 32 cases are shown in Table 6. The ASF group had significantly lower Risser grades and shorter instrumentation lengths than the PSF group. There were no significant differences in age at surgery and the number of vertebrae in the major curve between the two groups (Table 6). The time course changes of the radiographic parameters of cases with ≥32 mm of the preoperative LIV translation are shown in Figure 4. The preoperative TL/L curves were significantly larger in the ASF group than the PSF group (*p* < 0.01), but were equally improved postoperatively and at final follow-up (*p* = 0.35, *p* = 0.31, respectively). Similarly, the thoracic curves were larger in the ASF group preoperatively (*p* = 0.04), and were equivalent postoperatively and at final follow-up (*p* = 0.10, *p* = 0.17, respectively). While C7-CSVL before and immediately after surgery was the same in the two groups (*p* = 0.12, *p* = 0.29, respectively), C7-CSVL at the final follow-up was significantly larger in the PSF group than the ASF group (*p* = 0.02). Apical vertebral translation, which was the same preoperatively (*p* = 0.10), was significantly better corrected in the ASF group than the PSF group postoperatively and at the final follow-up (*p* = 0.03, *p* < 0.01, respectively). There was no significant difference in the L4 tilt between the groups at each time point (*p* = 0.44, *p* = 0.29, *p* = 0.41, respectively). Preoperative LIV translation from the CSVL was significantly larger in the ASF group than the PSF group (*p* = 0.02) but was significantly corrected in the ASF group postoperatively and at the final follow-up (*p* = 0.01, *p* = 0.03, respectively). The PSF group had a significantly larger LIV tilt at the final follow-up and a larger LIV disc angle before surgery than the ASF group (*p* < 0.01, *p* < 0.05, respectively).

Finally, a comparative analysis was performed for 30 ASF cases with LIV of L2 and that of L3 (Table 7). There was no significant difference between the groups at the final follow-up except for instrumentation length.

## 4. Discussion

LIV selection is thought to be an important factor affecting distal mobile segments and global imbalance in corrective surgery for Lenke 5 AIS [4,9]. However, a limited number of studies has focused on LIV translation and analyzed it according to surgical approaches, including PSF and ASF. We found that a large LIV translation from the CSVL at the final follow-up resulted in a larger L4 tilt and poorer global coronal balance. In patients with preoperative LIV translation ≥ 32 mm, LIV translation remained after PSF in some cases, causing insufficient LIV tilt correction and coronal plane imbalance. Furthermore, ASF demonstrated a better correction with shorter spinal segment fusion than PSF, even in cases with large preoperative LIV translation.

Regarding the LIV selection for the TL/L curve AIS surgery, previous reports suggested that a LIV below L4 was associated with the risk of intervertebral disc degeneration and back pain [10,11,12,13,14,15,16]. It is, thus, important to select the LIV to be the cephalad of L3 to maximize the lumbar motion segments [10,11,12,13,14,15,16]. Although several studies reported on the lumbar spinal level with respect to LIV selection [10,11,12,13,14,15,16], few reports have considered the various parameters of the LIV. Li et al. [17] reported that LIV tilt was an important radiological parameter correlated with postoperative global balance in PSF, and that failure to reduce the LIV tilt to <8° postoperatively suggested an increased risk of postoperative global coronal imbalance. Wang et al. [5] found a significant correlation between preoperative LIV translation and spinal balance 2 years postoperatively in Lenke 5 AIS cases (10 ASF and 20 PSF cases). Lastly, Phillips et al. [14] reported that preoperative LIV translation was an important predictor of success in 139 cases of Lenke 5 AIS surgery (including ASF and PSF) with L3 as the LIV. The present study demonstrated that a 12 mm LIV translation at the final follow-up could be the reference value for a good correction of the L4 tilt and global coronal balance. Furthermore, LIV translation was significantly correlated with various radiological parameters at the final follow-up. These findings suggest that LIV translation is an important factor in LIV selection.

Although it is known that LIV centering and leveling are important in creating a good global balance in Lenke 5 curve corrective surgery [14,17], it is unclear what preoperative LIV translation selection criteria will result in good balance. In a meta-analysis, Wang et al. [5] reported an overall preoperative LIV translation value of 28 mm, which can be considered as a general criterion for LIV selection. Phillips et al. [14] analyzed 2-year postoperative outcomes in 139 patients who underwent ASF or PSF surgery with L3 as the LIV. They reported that the preoperative LIV translation was an important predictor of ideal outcomes with respect to correction and alignment, with a LIV translation of < 35 mm being a potential threshold for LIV selection [14]. However, these reports did not evaluate the LIV translation according to different surgical approaches. In this study, we found that a preoperative LIV translation of 32 mm could be the reference value for the LIV selection in the posterior approach. However, no significant preoperative LIV translation cutoff values were identified for the anterior approach. The preoperative LIV translation of the ASF group was significantly larger than that of the PSF group; however, after the correction it was significantly smaller in the ASF group than the PSF group at the final follow-up. Anterior dual-rod instrumentation surgery for the TL/L curve can achieve remarkable three-dimensional and stable correction with a shorter fusion segment than a PSF surgery, which uses pedicle screws [11,12,26]. Furthermore, this study demonstrated that even in cases with a large preoperative LIV translation, ASF can centralize the LIV better than PSF and provide better main TL/L curve correction and global coronal balance. In cases with preoperative L3 translation > 32 mm, a PSF surgery with L3 as the LIV could result in residual postoperative LIV translation, resulting in remaining LIV tilt and coronal plane imbalance. In such cases, the ASF approach could be useful in obtaining good curve correction and global balance without fusion to L4.

The main limitation of this study relates to the postoperative follow-up period. Patients in the ASF group were followed for >10 years and those in the PSF group for >2 years. However, the postoperative course of ASF is reported to be stable over an extended follow-up of more than 10 years [11], making it an acceptable comparison to the PSF group with a 2-year follow-up regarding the initial correction. However, a long-term follow-up after a PSF surgery is necessary for long-term stability, possible symptoms, and disc degeneration below the fusion mass and distal adding on, especially in cases with residual LIV displacement. The other limitation is that there were no AIS cases that underwent PSF surgery with LIV to L4, and we cannot compare the results with those of cases that had LIV to L3, although LIV translation would become smaller when LIV is set to L4 than when it is set to L3.

## 5. Conclusions

In corrective surgery for Lenke 5 AIS, a large LIV translation from the CSVL at the final follow-up results in a larger L4 tilt and poorer global coronal balance. To obtain a good correction and maintenance of LIV tilt and coronal plane balance, a preoperative LIV translation of 32 mm could be a reference value for LIV selection in the posterior approach. ASF can centralize the LIV better than PSF with shorter spinal segment fusion and could be useful in obtaining a good curve correction and global balance without fixation to L4 in cases with large preoperative LIV translations.

## Figures and Tables

**Figure 1 jcm-12-01389-f001:**
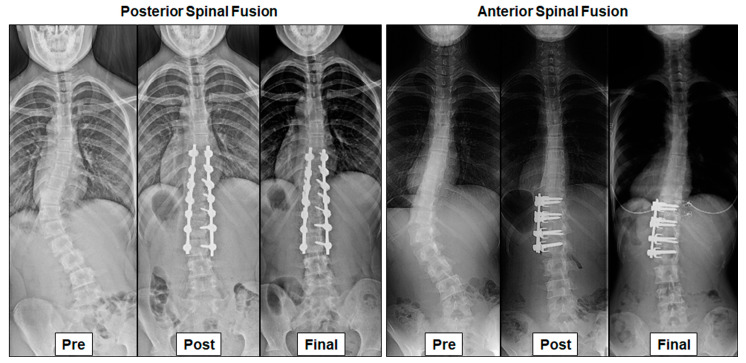
Radiographs of posterior and anterior spinal fusion for thoracolumbar adolescent idiopathic scoliosis at preoperative, postoperative, and final follow-up.

**Figure 2 jcm-12-01389-f002:**
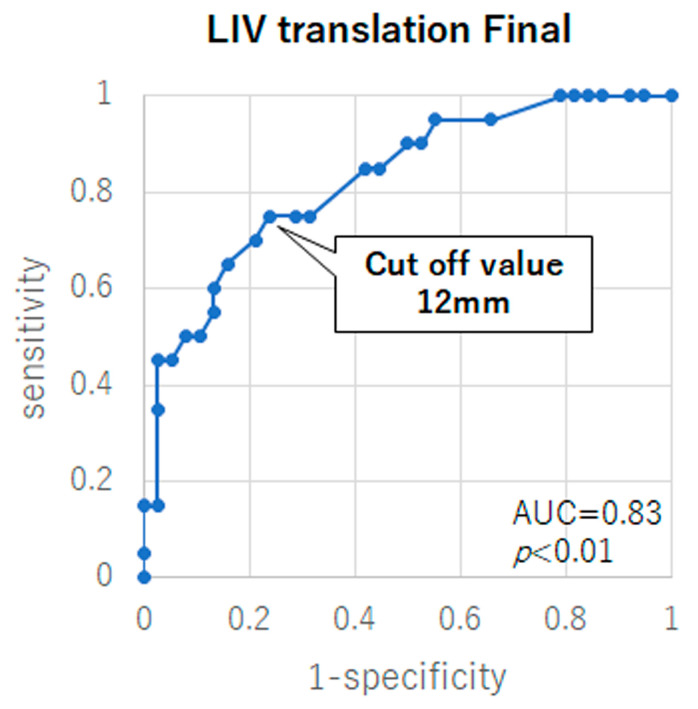
Receiver-operating characteristic (ROC) curve of the lowest instrumented vertebra (LIV) translation at the final follow-up for good outcomes (L4 tilt < 8° and C7 translation from the center sacral vertical line <15 mm) at the final follow-up. AUC indicates area under the ROC curve.

**Figure 3 jcm-12-01389-f003:**
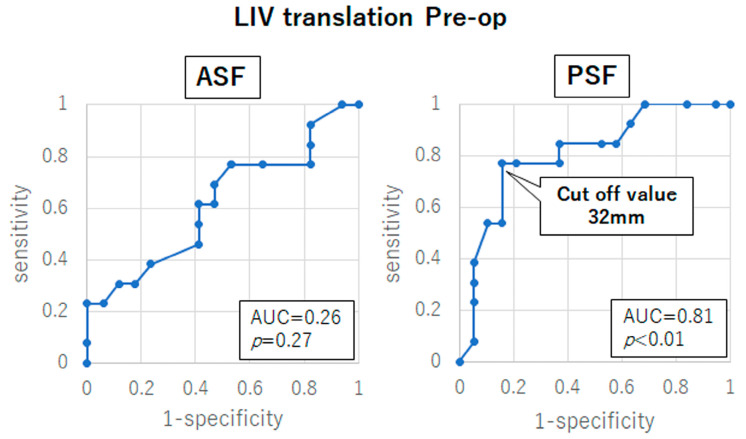
Receiver-operating characteristic (ROC) curve of the preoperative lowest instrumented vertebra (LIV) translation of ≤12 mm at the final follow-up in the anterior spinal fusion (ASF) and the posterior spinal fusion (PSF) groups. AUC indicates area under the ROC curve.

**Figure 4 jcm-12-01389-f004:**
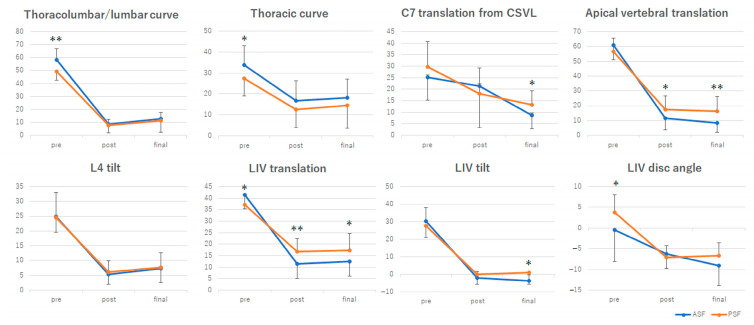
Time course changes in the radiographic parameters of the anterior spinal fusion (ASF) and posterior spinal fusion (PSF) groups in cases with preoperative lowest instrumented vertebra (LIV) translation ≥ 32 mm. CSVL indicates the center sacral vertical line. Error bars represent standard deviation. * *p* < 0.05, ** *p* < 0.01.

**Table 1 jcm-12-01389-t001:** Patient demographic data of ASF and PSF group.

	ASF (*n* = 30)	PSF (*n* = 32)	*p*-Value
Age at surgery (years)	14.4 ± 1.9	15.7 ± 2.0	<0.01 *
Risser sign	3.1 ± 0.8	4.0 ± 1.0	<0.01 *
Number of vertebrae in major curve (upper-end to lower-end vertebra)	4.8 ± 0.6	5.2 ± 0.8	0.01 *
LIV (vertebral level: cases)	L2: 13, L3: 17	L3: 32	
Instrumentation length (segments)	3.7 ± 0.7	5.4 ± 1.0	<0.01 *
Operation time (min.)	281.1 ± 47.0	205.7 ± 39.8	<0.01 *
Intraoperative blood loss (mL)	416.5 ± 234.6	435.7 ± 263.6	0.39
Follow-up periods (yrs)	17.2 ± 3.1	2.0 ± 0.1	<0.01 *

The values are given as the mean ± standard deviation. * Statistically significant. ASF indicates anterior spinal fusion; PSF, posterior spinal fusion, LIV, lowest instrumented vertebra.

**Table 2 jcm-12-01389-t002:** Radiographic parameters for all 62 cases.

	Preop.	Postop.	Final
Coronal plane data			
Thoracolumbar/lumbar curve (°)	51 ± 9	7 ± 6 *	10 ± 9 *
Thoracic curve (°)	29 ± 10	15 ± 9 *	16 ± 10 *
Sagittal plane data			
Thoracic kyphosis (T5-12) (°)	15 ± 8 *	21 ± 11 *	24 ± 12 *
Lumbar lordosis (L1-S1) (°)	45 ± 10	44 ± 11	51 ± 12 *
Balance parameters and translational data			
C7 translation from CSVL (mm)	22 ± 12	20 ± 14	10 ± 6 *
Sagittal vertical axis (mm)	19 ± 16	19 ± 16	22 ± 17
Apical vertebral translation (mm)	52 ± 12	13 ± 9 *	10 ± 9 *
L4 tilt (°)	22 ± 7	5 ± 4 *	6 ± 5 *
LIV translation from CSVL (mm)	33 ± 9	12 ± 6 *	12 ± 8 *
LIV tilt (°)	26 ± 8	−1 ± 5 *	−2 ± 6 *
Disc angle below LIV (°)	3 ± 6	−6 ± 3 *	−7 ± 4 *

The values are given as the mean ± standard deviation. * Statistically significant compared to preop (*p* < 0.05). ASF indicates anterior spinal fusion; PSF, posterior spinal fusion; CSVL, central sacral vertical line; LIV, lowest instrumented vertebra.

**Table 3 jcm-12-01389-t003:** Radiographic parameters of ASF and PSF group.

	ASF (*n* = 30)	PSF (*n* = 32)	*p*-Value
Thoracolumbar/lumbar curve			
Preoperatively (°)	56 ± 9	46 ± 7	<0.01 *
Flexibility (%)	75 ± 12	85 ± 16	<0.01 *
Immediately postoperative (°)	8 ± 6	6 ± 5	0.10
Final follow-up (°)	11 ± 8	9 ± 8	0.14
Correction rate at final follow-up (%)	80 ± 14	81 ± 14	0.36
Thoracic curve			
Preoperatively (°)	34 ± 9	25 ± 8	<0.01 *
Immediately postoperative (°)	17 ± 10	13 ± 8	0.02 *
Final follow-up (°)	19 ± 10	14 ± 9	0.01 *
Correction rate at final follow-up (%)	46 ± 23	51 ± 22	0.18
Thoracic kyphosis (T5-12) (°)			
Preoperatively (°)	17 ± 9	13 ± 8	0.03 *
Immediately postoperative (°)	28 ± 9	13 ± 7	<0.01 *
Final follow-up (°)	31 ± 10	16 ± 8	<0.01 *
Lumbar lordosis (L1-S1) (°)			
Preoperatively (°)	45 ± 10	46 ± 10	0.27
Immediately postoperative (°)	48 ± 9	41 ± 10	<0.01 *
Final follow-up (°)	54 ± 11	48 ± 11	0.03 *
C7 translation from CSVL (mm)			
Preoperatively	19 ± 12	25 ± 11	0.02 *
Immediately postoperative	21 ± 16	19 ± 12	0.27
Final follow-up	9 ± 6	10 ± 6	0.32
Sagittal vertical axis (mm)			
Preoperatively	17± 16	21 ± 17	0.18
Immediately postoperative	16 ± 15	21 ± 17	0.12
Final follow-up	17 ± 14	25 ± 16	0.03 *
Apical vertebral translation (mm)			
Preoperatively	54 ± 13	50 ± 11	0.08
Immediately postoperative	12 ± 8	14 ± 9	0.20
Final follow-up	9 ± 6	11 ± 11	0.11
L4 tilt			
Preoperatively (°)	23 ± 6	20 ± 7	0.04 *
Immediately postoperative (°)	5 ± 4	4 ± 4	0.25
Final follow-up (°)	7 ± 5	5 ± 5	0.12
LIV translation from CSVL (mm)			
Preoperatively	36 ± 10	30 ± 7	<0.01 *
Immediately postoperative	12 ± 7	13 ± 6	0.33
Final follow-up	12 ± 6	12 ± 9	0.39
LIV tilt (°)			
Preoperatively	27 ± 9	26 ± 7	0.24
Immediately postoperative	−2 ± 4	0 ± 6	0.09
Final follow-up	−4 ± 5	1 ± 7	<0.01 *
Disc angle below LIV (°)			
Preoperatively	0 ± 8	5 ± 4	<0.01 *
Immediately postoperative	−6 ± 3	−5 ± 3	0.11
Final follow-up	−9 ± 5	−5 ± 3	<0.01 *

The values are given as the mean ± standard deviation. * Statistically significant. ASF indicates anterior spinal fusion; PSF, posterior spinal fusion; CSVL, central sacral vertical line; LIV, lowest instrumented vertebra.

**Table 4 jcm-12-01389-t004:** Correlation analysis between LIV translation from CSVL and radiographic parameters at final follow-up.

	Correlation Coefficient	Statistical Significance
Thoracolumbar/Lumbar curve		
Correction rate	r = −0.64	*p* < 0.01 *
Correction loss	r = 0.42	*p* < 0.01 *
C7 translation from CSVL	r = 0.38	*p* < 0.01 *
Apical vertebral translation	r = 0.83	*p* < 0.01 *
L4 tilt	r = 0.69	*p* < 0.01 *
LIV tilt	r = 0.47	*p* < 0.01 *
Disc angle below LIV (°)	r = −0.21	*p* = 0.10

* Statistically significant. LIV indicates lowest instrumented vertebra; CSVL, central sacral vertical line.

**Table 5 jcm-12-01389-t005:** Demographic and radiographic parameters at final follow-up.

	L4 Tilt < 8° and C7-CSVL < 15 mm (*n* = 38)	L4 Tilt ≥ 8° or C7-CSVL ≥ 15 mm (*n* = 24)	*p*-Value
Age at surgery (yrs)	14.9 ± 2.0	15.4 ± 2.1	0.18
Risser sign	3.5 ± 1.0	3.7 ± 1.1	0.27
Number of vertebrae in major curve (upper-end to lower-end vertebra)	5.1 ± 0.7	5.0 ± 0.8	0.32
Surgical procedure	ASF: 19, PSF: 19	ASF:11, PSF:13	0.38
LIV (vertebral level: cases)	L2: 8, L3: 30	L2: 5, L3: 19	
Instrumentation length (segments)	4.6 ± 1.2	4.5 ± 1.3	0.30
Thoracolumbar/lumbar curve (°)	7 ± 6	15 ± 9	<0.01 *
Thoracic curve (°)	16 ± 9	17 ± 11	0.31
Thoracic kyphosis (T5-12) (°)	26 ± 13	21 ± 11	0.07
Lumbar lordosis (L1-S1) (°)	52 ± 12	50 ± 11	0.24
C7 translation from CSVL (mm)	7 ± 4	14 ± 6	<0.01 *
Apical vertebral translation (mm)	6 ± 7	16 ± 7	<0.01 *
L4 tilt (°)	3 ± 3	10 ± 5	<0.01 *
Sagittal vertical axis (mm)	17 ± 13	27 ± 17	<0.01 *
LIV translation from CSVL (mm)	8 ± 6	17 ± 6	<0.01 *
LIV tilt (°)	−4 ± 5	2 ± 7	<0.01 *
Disc angle below LIV (°)	−6 ± 3	−8 ± 6	0.10

The values are given as the mean ± standard deviation. * Statistically significant. CSVL indicates central sacral vertical line; ASF, anterior spinal fusion; PSF, posterior spinal fusion; LIV, lowest instrumented vertebra.

**Table 6 jcm-12-01389-t006:** Demographic data of cases with ≥ 32 mm of preoperative LIV translation from CSVL.

	ASF (*n* = 19)	PSF (*n* = 13)	*p*-Value
Age at surgery (years)	14.5 ± 2.1	14.9 ± 1.6	0.26
Risser sign	3.2 ± 0.8	3.8 ± 0.7	0.01 *
Number of vertebrae in major curve (upper-end to lower-end vertebra)	4.9 ± 0.6	5.0 ± 0.8	0.41
LIV (vertebral level: cases)	L2: 11, L3: 8	L3: 13	
Instrumentation length (segments)	3.8 ± 0.5	5.2 ± 0.9	<0.01 *

The values are given as the mean ± standard deviation. * Statistically significant. LIV indicates lowest instrumented vertebra; CSVL, central sacral vertical line; ASF, anterior spinal fusion; PSF, posterior spinal fusion.

**Table 7 jcm-12-01389-t007:** Demographic and radiographic data of ASF cases with LIV of L2 and L3.

	LIV: L2 (*n* = 13)	LIV: L3 (*n* = 17)	*p*-Value
Number of vertebrae in major curve (upper-end to lower-end vertebra)	4.9 ± 0.7	4.8 ± 0.5	0.36
Instrumentation length (segments)	3.4 ± 0.6	3.9 ± 0.7	0.03 *
Thoracolumbar/lumbar curve (°)	12 ± 5	11 ± 9	0.42
C7 translation from CSVL (mm)	9 ± 6	10 ± 6	0.31
Apical vertebral translation (mm)	9 ± 7	8 ± 5	0.31
L4 tilt (°)	8 ± 5	6 ± 5	0.13
Sagittal vertical axis (mm)	17 ± 14	16 ± 13	0.47
LIV translation from CSVL (mm)	12 ± 7	12 ± 6	0.50
LIV tilt (°)	−4 ± 4	−4 ± 5	0.39
Disc angle below LIV (°)	−9 ± 4	−9 ± 5	0.43

The values are given as the mean ± standard deviation. * Statistically significant. ASF indicates anterior spinal fusion; CSVL, central sacral vertical line; LIV, lowest instrumented vertebra.

## Data Availability

The data that support the findings of this study are available from the corresponding author on reasonable request.

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
