# Peer review of "Influence of Lateral Translation of Lowest Instrumented Vertebra on L4 Tilt and Coronal Balance for Thoracolumbar and Lumbar Curves in Adolescent Idiopathic Scoliosis"

_jcm, 2023, doi:10.3390/jcm12041389_

Round 1

Reviewer 1 Report

The paper is interesting, even if it is not an absolute novelty. I would ask for greater attention in the explanation of the radiographic measurement technique, possibly adding explanatory Rx figures before and after surgery, of the two techniques (anterior and posterior instrumented correction and arthrodesis). I noticed that on page 2, lines 85,86, 87 and 92,93, the technique for measuring the tilt of L4 and LIV is written incorrectly. The inclination of the upper  endplate is measured only for the most proximally instrumented vertebra.

I enclose an extract from the Radiographic Measurements Manual of the group of L.G. Lenke

Reviewer 2 Report

Thank you for submitting your study on the LIV in Lenke et al type 5 AIS curves. There has been a long-standing debate on whether the LIV can safely be at L3 or perhaps higher rather than L4 or lower. Your data indicates that it is safe to have an LIV at L3 if the preoperative lateral translation is 32 mm or less at that level. As a consequence, I feel your study has potential for publication. Unfortunately, I have numerous questions to be addressed before a final decision can be made. These include:

(1). Overall, I felt your manuscript was difficult to read and comprehend. I read it on several occasions in preparation for this report. In general, please remember that Cobb is not an angle but rather a technique used to measure spinal angles (scoliosis, kyphosis, lordosis, etc). Cobb angle is a common term, but it is jargon. The correct term for scoliosis is major coronal curve or just major curve. Also, decimal places are not commonly used in radiographic spinal measurements as the standard error of measurement is 3-5 degrees. Therefore, decimal places do not add accuracy or significance. Please round to the nearest whole number. The statistics will need to be recalculated following the rounding. These might change owing to the small differences in some of the figures. It should also make the results easier for the readers to comprehend. 

(2). Although there were 13 patients with an LIV of L2 and 17 patients to L3 with ASF their results were combined. It would have been an interesting comparison if they were also reported separately.

(3). There were no PSF patients with an LIV to L4 as all 32 patients had an LIV to L3. This would have been a very interesting comparison group.

(4). The follow-up between groups was very disparate as it was 17.2 +/- 3.1 years in ASF group compared to only 2.0+/- 0.1 years in the PSF group. Since this study is basically dealing with long-term follow-up results are these comparable, particularly with initial correction as well as long-term stability and possible symptoms? It appears that you have had treatment preference change despite similar results. Why? I have always thought that ASF yielded better correction because of the increased apical mobility induced by anterior discectomies. It also preserved distal motion segments.

Round 2

Reviewer 2 Report

Thank you for revising your manuscript. It has been significantly improved and is now ready for publication. It will be an interesting addition to the literature on this topic.